

# Brief Communication: Mapping river ice using drones and structure from motion

Knut Alfredsen[1], Christian Haas[2], Jeffrey A. Tuhtan[3], Peggy Zinke[1]

[1]Department of Civil and Environmental Engineering, Norwegian University of Science and Technology, 7491 Trondheim,
Norway
[2]I AM HYDRO GmbH, Märtishofweg 2, 78112 St. Georgen, Germany
[3]Centre for Biorobotics, Tallinn University of Technology, Estonia

*Correspondence to*: Knut Alfredsen (knut.alfredsen@ntnu.no)

**Abstract.** In cold climate regions, the formation and break-up of river ice is important for river morphology, winter water supply, riparian and instream ecology as well as for hydraulic engineering. Data on river ice is therefore significant, both to understand river ice processes directly and to assess ice effects on other systems. Ice measurement is complicated due to difficult site access, the inherent complexity of ice formations and the potential danger involved in carrying out on-ice measurements. Remote sensing methods are therefore highly useful, and satellite imagery, data from satellite-based radars as increasingly aerial and terrestrial imagery are currently applied. Access to low cost drone systems with quality cameras and Structure from Motion software opens up a new possibility for mapping complex ice formations. Through this method, a georeferenced DEM can be built and data on ice thickness, spatial distribution and volume can be extracted without accessing the ice and with considerable less measurement efforts compared to traditional surveying methods. The methodology applied to ice mapping is outlined here, and examples are shown on how to successfully apply the method to derive data on ice processes.

## 1 Introduction

River ice is a critical part of the cryosphere (Brooks et al., 2013), and ice formation has implications for river geomorphology, instream and riparian ecology, winter water supplies and for hydraulic engineering in cold climates. Physical monitoring and mapping ice is methodologically challenging, as access to river ice is wrought with difficult and potentially dangerous. Therefore, remote sensing methods are applied to assess ice in rivers and a number of examples of this exist in literature. Chu and Lindenschmidt (2016) integrated satellite images (MODIS) with radar data (RADARSAT-2) to assess freeze-up, break-up and ice types in the Slave River, Canada. They also evaluated the dataset with aerial- and terrestrial imagery to evaluate the satellite data, and found good agreement on break-up and freeze-up dates. Mermoz et al. (2014) demonstrated how ice thickness could be derived from RADARSAT-2 data for three rivers in Canada. Beltaos and Kääb (2014) used successive satellite imagery to assess the flow velocity and discharge during ice break-up, illustrating the ability of using remote sensing to study ice processes dynamics. Ansari et al. (2017) developed algorithms to automatically derive ice phenology data from bankside



photography, and produced time series of these ice data for the lower Nelson River, Canada. Most satellite-based approaches described in the existing literature are applied to large rivers and may not be applicable to smaller rivers and streams due to the resolution of the satellite imagery. Methods are therefore needed to monitor ice formation in smaller rivers. Further work is also needed to move from a qualitative evaluation of ice (focussing on ice types and presence or non-presence of ice) towards
a quantification of ice volumes and the spatial distribution of ice.

These gaps may be filled by use of  inexpensive aerial drones. Such drones increasingly have camera systems of sufficient quality, and the resultant aerial imagery can be processed using the Structure from Motion (SfM) method (Westoby et al. 2012) into a three-dimensional surface representation of the area covered by imaged area. Combined with ground control points,
surface models can be georeferenced and combined with other spatial data for analysis. Based on the georeferenced point cloud and aerial images, digital elevation models of high accuracy and resolution can be developed. Additionally, derived surface texture can be overlaid onto surface models and imagery can be undistorted and reprojected as georeferenced orthophoto mosaics to provide an accurate aerial image of the study object. SfM has shown a large potential within geosciences where detailed digital models are often needed (Westoby et al., 2012;Smith et al., 2016), where the method has been applied, for
example, to studies on river habitat and hydromorphology (Woodget et al., 2017), erosion and sedimentation studies (Smith and Vericat, 2015) and for grain size classification (Vasquez-Tarrio et al., 2017;Arif et al., 2016). Within the cryospheric sciences, the method has been used in the study of glaciers e.g (Ryan et al., 2015) and in the study of snow accumulation (Nolan et al., 2015).
Combining a drone with SfM may be an efficient tool for mapping ice in rivers of various sizes, with the added advantage of
also being able to cover small streams. This approach may enable detection of ice jams forming during ice runs and anchor ice dams forming during freeze up. These dynamic ice forms are difficult and time consuming (sometimes impossible) to map using traditional methods, but are important to many ice assessments. Ice break-up and associated jams can cause erosion and flooding and thereby severe damage to infrastructure and the riverine flora and fauna (Beltaos, 1995;Prowse and Culp, 2003). Anchor ice dams controls the freeze–up process in small rivers and streams, and are thereby important for understanding winter
conditions in such streams (Stickler et al., 2010;Turcotte and Morse, 2011)

In this brief communication, we outline the process of mapping river ice with a common, consumer-grade drone combined with commercial SfM software. Furthermore, we use the method to map a stranded ice jam and a river section with anchor ice dams and show examples of output of the method. Cross sections and volume of the ice run are computed from the mapped
data, and the location and size of the anchor ice dams are estimated. Since these quantitative data are not commonly produced due to the difficulties associated with traditional field measurements, the application of the method presented has the potential to increase the amount and quality of data available and further improve our understanding of ice processes that, form the foundation for modelling and practical assessments of ice impacts to built and natural environments.



## 2 Materials and methods

### 2.1 Study site

Our two study sites are located at the river Gaula (63.06°N, 10.29°E) and the tributary Sokna (62.98°N, 10.22°E) south of Trondheim, Norway. We mapped a stranded part of an ice run that took place in December 2016 at the Gaula site close to

Haga bru and at the Sokna site we mapped a section of the river with several broken anchor ice dams. The drone flight was carried out in February of 2017 under sparsely cloudy conditions with low wind velocities. The length of the reaches mapped are 350 m and 200 m for Gaula and Sokna respectively. During the measurements, the discharge in Gaula (at gauge Gaulfoss) was approximately 20 $m^3s^{-1}$ and in Sokna (at gauge Hugdal bru) approximately 3.5 $m^3s^{-1}$. This correspond to 28% of the annual mean annual flow.

### 2.2 Data collection and processing

At the two sites we acquired aerial images using a DJI Phantom3 Professional drone (www.dji.com), and surveyed ground control points (GCPs). The drone has a GPS/GLONASS assisted hover function and the 12 megapixel, on-board camera has a f/2.8 94-degree field of view and is mounted with a three-axis gimbal. The flight level was set to 30 m above the ground, but the level varies during the flight due to wind, pressure changes and the automated hover function. Most pictures were taken in

plan view, but some oblique pictures were taken to better capture vertical features of the ice formations. A minimal image overlap of 20% was used to ensure good picture alignment in the SfM analysis. For both sites, the drone was manually controlled from the ground using the DJI remote controller and an Apple iPad with the DJI Go application for flight control. Though capable of autonomous operation according to a pre-planned flight plan with the DJI GS Pro application, the drone was controlled manually due to the size of the study area and regulations on UAV operation. Each picture taken is stored on

board the drone with its associated Exif image information and the GPS position. A total of 215 and 68 pictures were taken of the Gaula and Sokna sites, respectively. However more frames were taken to increase the number of cameras as input for post processing as light conditions were poor on both sites because of cloud cover and low sun altitude in combination with steep valleys. The time spent in Gaula was about 15 minutes for rigging and then 17 minutes for acquiring the imagery. At the Sokna site, the drone was already rigged and 10 minutes was used for the flight. All GCPs and post-processing control points were

imaged during the drone flights. The GCPs and post- processing control points were surveyed using a Leica Viva RTK-GPS (spatial accuracy of 1-2 cm), and they were identified using custom-made numbered markers (25x25 cm) made of heavy red tarpaulin that were clearly visible on the drone images. At the Gaula site eleven ground control points (GCPs) were measured on the ice surface for georeferencing, and another 9 points were measured to provide a simple control of accuracy of the DEM after processing. At the Sokna site, 9 GCPs were measured on the river bank since it was not possible to access the ice surface

for safety reasons. The captured images and GCPs was processed using Agisoft Photoscan Pro version 1.3.0 (www.agisoft.com) into a georeferenced point cloud using the SfM workflow described by the following steps:



1. Photos were aligned and images with a quality index less than 0.8 were not included in the point cloud generation. The camera models were optimized and points with a RMS re-projection error of less than 0.2 were deleted from the cloud. Afterwards, the modelled camera positions and orientations were optimized again to improve the initial camera alignments, which were estimated by the software based on common features between pairs of images.

2. GCPs and control points were imported to Photoscan and markers are manually identified and linked to the GCP. Each marker was identified on a minimum of three images before the georeferencing parameters were recalculated. In the Sokna case, one of the GCP showed an error that was considered too large, and this point was removed. The camera models were optimized once more and georeferenced clouds were generated.

3. The georeferenced point clouds are densified and the ground points are classified to create meshes. The meshes were then used to build DEMs and generate orthophoto mosaics.

Figure 1 shows the locations of the camera positions and the GCPs, the generated point cloud, the DEM and an orthophoto mosaic for the Gaula site. To extract cross sections and analyse the DEM generated in Photoscan, data was exported to ArcMap 10.5 (www.esri.com), and the tools in the 3D analyst package was used to extract data from the DEM. The ground elevation under the ice jam in Gaula were taken from the 1 meter Norwegian DEM provided by the Norwegian mapping agency (www.hoydedata.no).

## 3 Results

The quality of the georeferencing and post-processing control points was assessed using the Root Mean Square Error (RMSE) of the combined x,y,z coordinates computed in Photoscan, and the results are shown in Table 1. The accuracy of the digital elevation model is considered good, and a further comparison of the georeferenced images with features on the digital map of the area show good correspondence between the features georeferenced in the images and the same features on the digital maps.

Figure 1(d) shows the orthophoto mosaic of the Gaula site, and the extent of the ice jam and the directional stacking of the ice floes accumulated in the jam. The formation of the ice cover on the river section outside the ice jam is also visible, and the pattern with retrograde build-up of ice with the leading edge is clearly identifiable in the lower left edge of the picture. Furthermore, cracking and some shoving of the ice cover outside the ice jam can be seen, giving a clear indication of the mechanisms that forms the ice cover on this reach of the river.

As shown in Figure 2, cross-sections of ice jams were extracted from a combination of the SfM- derived DEM and ground elevation data from the 1 m resolution Norwegian national DEM. Based on visual observations after the ice event and from observations in the field the jam was assumed to be grounded for most of its area. At the Gaula site, the mean thicknesses of the ice layers were varied from 2.34, 2.07 and 1.80 m for cross sections A, B and C, respectively, and the sheer wall heights



was 1.12, 0.92 and 1.74 m in cross sections A, B and C, respectively. The mean thickness of the jam was 2.2 m and the maximum thickness was 3.55 m. The ice jam volume was computed to be 6805 m³, covering a surface area of 3014 m². This is most likely a minimum volume since it is difficult to determine if the outermost meters of the DEM describing the ground is the actual ground level or the water surface of the river. The DEM used does not cover terrain under water, and it is not clear

if we see the bottom or the water surface in the outer part of the ground level plot on figure 2.

Figure 3 shows a number of broken anchor ice dams at the Sokna site. Dam locations and computed elevations and sizes can be used to derive how the ice cover will develop in the reach. This forms the foundation of understanding the hydraulic impacts of ice in steep rivers, and results can be combined with measured hydraulic variables for further evaluation. As an example,

Figure 3 shows two vertical profiles of a broken anchor ice dam where the elevations of dam crests over the water surface is measured to 0.8 and 1 meter respectively indicating the elevated water level in the reach when the dams were complete.

## 4 Discussion and conclusion

An example of using a simple drone combined with SfM to map river ice is shown for two important types of ice, an ice jam and anchor ice dams. The method outlined allows mapping of larger areas in small streams which is typically very difficult to

achieve with other remote sensing tools, and often impossible using lower-resolution satellite data products. The approach presented here using the drone is also significantly easier and more cost effective than using a fixed wing aircraft or a helicopter. Furthermore, the drone is also a more viable method were vegetation or valley size makes other aerial platforms difficult to use. Using a relatively small number of GPS measured ground points the point cloud was georeferenced, and a digital elevation model and a georeferenced orthophoto mosaic developed. Based on the georeferenced models, data on ice thickness, volume

and spatial distribution was extracted. Compared to previous measurements using GPS or total stations for mapping detailed spatial ice formations (e.g. Timalsina (2014)), the approach presented here removes the need to access the ice which in the Sokna case would have been difficult or impossible. Even if access to the ice was possible our new approach provides an amount and quality of data that cannot be matched by difficult and time consuming collection of manual measurements.

A challenge with remote sensing is to assess ice thickness over flowing water, and this is also the case using the drone and SfM method. As is in the case with the Gaula ice jam, it is relatively simple to compute ice volumes and derive ice cross sections for ice grounded on a surface with known ice-free geometry. In cases where river ice has free water the problem is more complicated, as is illustrated by considering the conditions shown in the Sokna case (Figure 3). Anchor ice dam positions and their widths and heights can be derived from the picture and the DEM, but unlike the Gaula case, no ice-free geometry of

the river is available to assess the volume of ice. A possible approach as we show is to derive the elevation of the level of the open water from the SfM generated DEM and then use this to assess the thickness of the anchor ice dams seen in the picture. This method has its limitations due to the turbulent surface of the water, and because the rocks and outcroppings known to be



important for anchor ice dam formation are not shown in the images captured after the ice is formed. To facilitate this approach, a flight during ice free conditions or some other kind of measurement campaign is needed to capture the size and position of large boulders and other morphological conditions that control placement of anchor ice dams. Together with the dam positioning methodology described herein, this approach could be used to accurately determine the favourable conditions for

development of anchor ice dams. The formation of anchor ice in Sokna and other similar small rivers is crucial for the formation of an ice cover and for the in-stream flow conditions (Stickler et al., 2010), therefore understanding the mechanisms for formation and the location of the ice would make analysis of winter conditions more precise and would improve the ability to predict dam positions.

Our experience suggests that the ice cover was better detected and represented under snow free conditions than from a comparable surface with an undisturbed snow cover. It is therefore advisable to fly the reach as soon as possible after an ice event has occurred and preferably before a snow covers develops.

In general, the DJI Phantom 3 Professional is limited in poor lighting conditions due to the small 6.17/55 mm sensor
(www.dji.com)- Compared to bigger sensors, the influence of lighting conditions can be drastic as shutter speed is decreased, with a commensurate increase of the aperture. Because of this, the camera system is more sensitive to the influence of vibrations and movement of the UAV caused by wind. Further, bright light with dark shadows can result in poor picture quality as the camera sensor can overload by bright reflections from unshaded areas, whereas shaded areas may lack suitable information for SfM applications. Therefore, we recommend manual camera settings on the Phantom 3 Professional to ensure
sufficient image quality for post processing. Automatic waypoint navigation (e.g. in this case with the DJI GS Pro application) can help to create a dataset with good coverage of the site and sufficient image overlapping. However, all autonomous navigation and auto camera settings increase the risk of poor picture quality, especially under less than optimal lighting conditions. Regardless of the approach used, data quality must be checked in the field to prevent data gaps caused by poor image acquisition.


Labelled regions were manually extracted for corresponding physical features (e.g. "cracks", "anchor ice") following generation of the orthopoto mosaic. However, processing algorithms for advanced pixel and object-based image classification have been used by the remote sensing community for decades, and geoscience practitioners have already begun to develop new methods specifically tailored to 3D point clouds derived from SfM models (Brostow et al., 2008). Once workflows for
the identification and classification of physical features specific to ice processes and types have been established, such as presented herein, it can reasonably be expected that machine learning may be applied to more efficiently classify large quantities of orthophoto imagery.

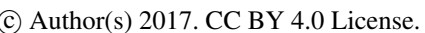


Assessment of river ice using a drone and SfM is a low cost and efficient method that is particularly suitable for small and medium sized rivers and streams stands out as an attractive option compared to other methods. Using satellites is difficult in smaller watersheds due to the small size of the rivers compared to satellite pixels, and the data can be costly. Using SfM with other aerial platforms like planes or helicopters is more expensive to operate, and are also not as versatile as a drone in narrow

valleys and at sites with heavy bankside vegetation. Manual measurements take far more time if similar detail is needed (and may not be possible in practise), and can be difficult due to necessary safety restrictions related to work on ice covered rivers. Thought no photographic images are produced, the use of a terrestrial laser scanner is a possible bankside remote sensing alternative (Wall et al., 2017) For larger or longer reaches, the number of scan positions occupied may lead to a more extensive field campaign, and further processing and integration of point clouds from many positions may also consume time and effort.

The approach presented here is applicable to several issues in river ice research, modelling and management. As seen in Figure 1 (d), the mechanisms of ice formation can be derived from the pictures, and since the method is not very time consuming, repeating measurements over a short time is viable and could provide detailed data of the freeze-up process. When combined with measurements of climate and hydraulic variables, this approach could benefit process understanding and model

development and also the calibration and validation process. Mapping of ice jams could provide a basis for the assessment of ice volumes and the configuration of floes and ice elements that make up the jam. These data are critical for understanding the river ice breakup processes and may aid in assessing potential damages that can result.

**Author contributions**

All authors contributed to the design of the field work based on previous measurements by KA. All authors participated in the

field work, and CH flew the drone and acquired the images. KA did the analysis supported by JAT and CH, and PZ provided the ice-free geometry data. KA wrote the manuscript with input from all co-authors.

**Acknowledgements**

JAT and CH participated in the field work as part of the Workshop on data processing from images gained by Unmanned Aircraft Systems that was funded by the European Economic Area (EEA) Grant "Restoration of the aquatic and terrestrial

ecosystems at Fundu Mare Island" (Project Number RO02-0008) with financial contributions from Norway and Romania.

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



Table 1 Root Mean Square Error computed from GCP and Control points. The error is reported for the combination of x, y, z and error for each coordinate.

| Measurement | # of points | Mean RMS Combined (m) | Min RMSE Combined (m) | Max RMS Combined (m) | Lon (m) | Lat (m) | Alt (m) |
|---|---|---|---|---|---|---|---|
| Gaula GCP | 11 | 0.072 | 0.023 | 0.098 | 0.050 | 0.050 | 0.013 |
| Gaula Control | 9 | 0.099 | 0.007 | 0.222 | 0.049 | 0.056 | 0.065 |
| Sokna GCP | 9 | 0.06 | 0.012 | 0.102 | 0.047 | 0.031 | 0.022 |





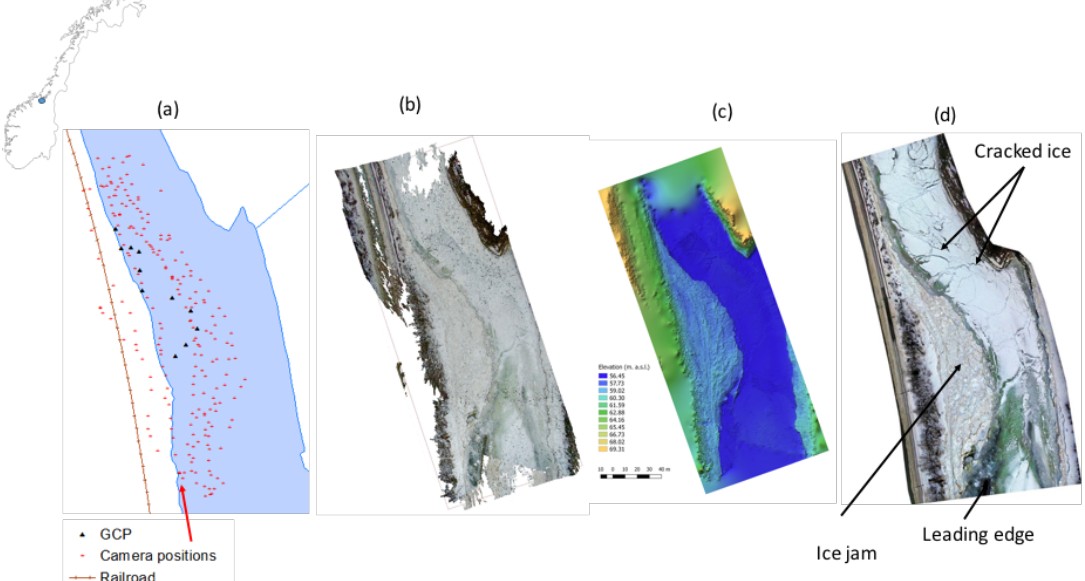

Figure 1 Stages of the process of mapping the Gaula site. Panel (a) shows the camera positions (crosses) and ground control points (triangles)
5  overlaid on a digital map of the river section. Panel (b) shows the dense point cloud for the same reach after processing the aerial imagery.
Panel (c) shows the georeferenced digital elevation model based on the point cloud. Panel (d) shows an orthophoto mosaic of the reach,
showing both the ice jam to the left and an ice cover at different stages of formation.





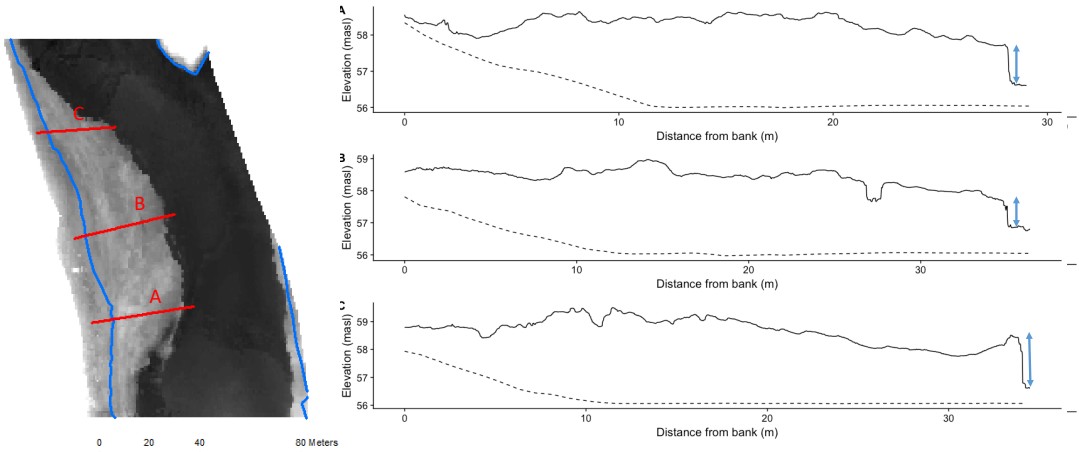

Figure 2 Cross sections through the Gaula ice jam at three different locations marked A, B and C. The height of the shear wall of the ice jam is marked with an arrow in each cross section. The dashed line indicates the ground level from the 1 meter resolution Norwegian DEM. Most of the jam is assumed to be grounded, but for the last 10 m it is uncertain if the DEM record the water level or the ground level.





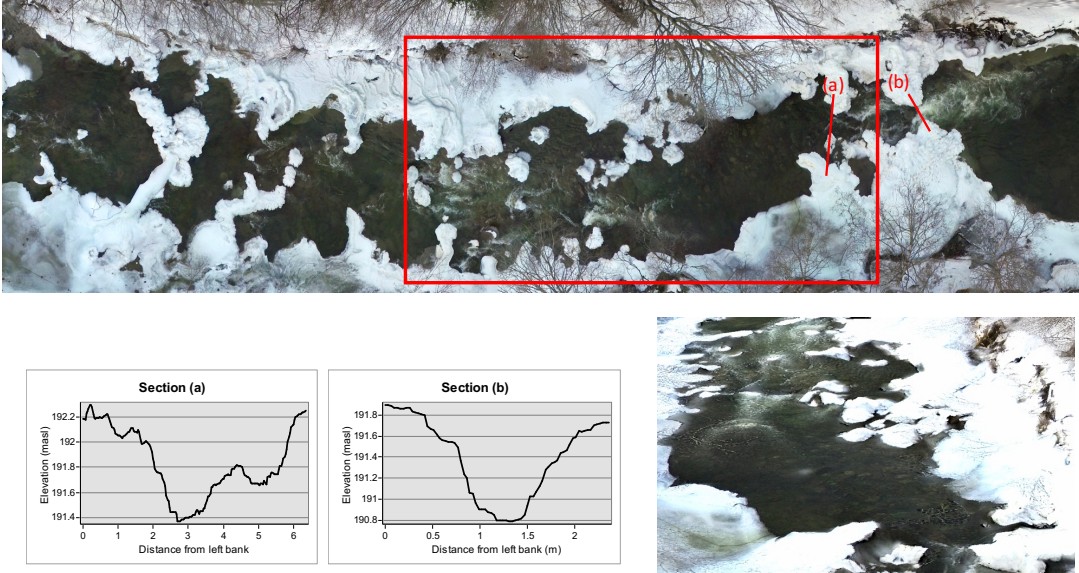

Figure 3 Partly broken anchor ice dams in the Sokna reach. The upper picture shows an orthophoto mosaic of the reach, the lower pictures shows and the two sections (a) and (b) marked on the picture and the area in the red rectangle seen from downstream.