# Peer review of "Brief Communication: Mapping river ice using drones and structure from motion"

_The Cryosphere, 2017_

## Referee Comment (RC1) · K.-E. Lindenschmidt (Referee) · 6 Nov 2017

This communique highlights the application of drones in river ice work. Such applications are becoming more and more essential as advances in river ice research are requiring more detailed descriptions of the characteristics and geomorphological settings of ice, hence this note is timely. It introduces a method to other river ice researchers seeking an inexpensive and safe alternative for ice cover mapping.

Before this communique is accepted for publication, some amendments and revisions are required:

Limitations of the photography are extensively elucidated (particularly on Page 6), however little is said about the limitations of using UAVs themselves. Examples may in-

clude: - additional qualifications and registration required of the UAV operator, - operations feasible only on wind-calm days, - only short flights are possible (also through regulations) and - UAVs cover smaller aerial extents compared to other methods. Perhaps these can be listed in the Introduction so that the reader can assess if these limitations would hinder their particular case studies.

The section of consolidated ice shown in Figure 1 is labelled 'ice jam'. This may not be quite correct since ice jams usually extend across the river width to cause backwater staging. Perhaps labelling this section as 'consolidated ice' or 'ice-jam remnant' is be more accurate.

Line 27 on Page 5 refers to "free water". I'm not sure what is meant here. "Open water" doesn't quite fit either since how can river ice have open water. Please restructure the sentence to clarify its meaning.

Some minor, editorial revisions include: Page 1, Line 23: replace "difficult and potentially dangerous" to "difficulties and dangers" Page 1, Line 30: "ice processes dynamics" should read "ice process dynamics" or "the dynamics of ice processes". Page 2, Line 33: perhaps replace "built" with "constructed" Page 3, Line 30: "were", not "was" Page 5, Line 17: "where", not "were"

---

## Referee Comment (RC2) · M. Loewen (Referee) · 10 Nov 2017

This is a timely and interesting paper describing an efficient and economical method for using inexpensive aerial drones to map river ice. I am quite certain that this method will prove to be a valuable new tool for river ice researchers. The method does not appear to be novel but its application to mapping of river ice is new and innovative. Accurate estimates of the aerial extent of various ice types can be made and under the right circumstances estimates of ice volumes are possible. The authors point out that the method allows for repetitive measurements to be made of the same reach because data can be acquired quickly and relatively easily. This will allow researchers to study the evolution of ice covers and the associated processes in much more detail than was previously possible.

The authors could consider discussing the following questions: How does the limited range of these inexpensive drones impact the applicability of this method to large rivers? I have deployed a similar drone on the Peace River, Canada and the width of the river exceeded the range in some places.

In extremely cold weather battery life can drop by 50%. Did the authors encounter any difficulties because of this effect?

Would a higher resolution camera improve the accuracy of this method?

The paper is well written and the topic will be of interest to readers of the journal. I recommend publication after the authors have addressed the minor revisions suggested below.

Specific Comments:

Page 1-Line 23: Delete "wrought with".

Page 2- Line 29: "…and show examples of output of the method." Awkward wording.

Section 2.1: Please provide more complete descriptions of the study reaches e.g. widths, slopes, geomorphology etc.

Page 3-Lines 21-23: Unclear - there is only one camera on the drone so increasing the number of cameras is clearly done in the software. Please explain this more fully.

Page 3-Lines 28-29: Are the 9 points referred to here called Control Points in Table 1? This is unclear.

Page 4-Line 1: Should this read "… index greater than 0.8…". Seems odd that when a quality index is greater than 0.8 that images were excluded. If space permits a brief explanation of how this quality index is computed would be helpful.

Page 4-Lines18-21: The authors write that "...the digital elevation model is considered good…". They are referring to the errors listed in Table 1 but a brief discussion of how

they arrived at this conclusion would be helpful to readers who are not familiar with DEM's.

Page 4-Line 33: Delete "varied from".

Page 5-Line 1: Change to "…were 1.12…".

Page 5-Lines 2-5: The meaning of "outermost" and "outer part" are a bit unclear.

Page 5-Line 22-23: Why was access to the Sokna reach difficult or impossible? Is it difficult to get to the stream or is it the ice conditions that make it unsafe?

Page 5-Lines 30-33 & Page 6-Lines 1-3: It is not clear to me how determining the open water elevation helps to assess the thickness of anchor ice dams. Anchor ice dams are anchored either to the bed or large rocks so how is their thickness related to the open water elevation? Perhaps I am missing something here but the middle section of this paragraph confused me.

Page 7=Lines 1-6: Much of this is repetitive. Perhaps it could be deleted?

Page 7-Lines 14-15: This is unclear, "…could benefit process understanding and model development and also the calibration and validation process." Please clarify.

Figure 1: Unclear, is camera position the location of the drone when a single image was taken? What is the red arrow in the lower left corner? If it is flow direction, please relocate it and label it or refer to it in the caption. Note that flow direction should be indicated in all figures.

Figure 2: Label the plots A, B and C. Where are the last 10 m in the plots? Or perhaps the last sentence of the caption can be deleted since I think this information should be in the text.

Figure 3: The last sentence in the caption has some grammatical errors.

[Figure]

---

## Referee Comment (RC3) · S. Clark (Referee) · 11 Nov 2017

The authors have presented an interesting paper regarding the use of unmanned aerial vehicles to map river ice extent and ice properties such as ice thickness and volume. It has also been our group's experience that UAVs can be very helpful for this purpose, as well as to quantify surface ice concentration and ice pan velocities. Their methodology should be useful for many other researchers in the field as well. I have a few minor comments, followed by some grammatical suggestions. It would have been helpful to have additional detail regarding the specifics of what constitutes a high quality image vs a low quality image, and how one might be more likely to achieve the former. Any practical tips for a successful flight would be appreciated by the readers. It also would have been nice to see whether the chosen number of control points was actually nec-

essary. Is it a coincidence that the number of points only varied between 9 and 11? Would the error have been significantly increased if only 5 control points had been used? Are there any suggestions for the placement of these points? For instance, had the opposite bank been easily accessible, would it have been better to have the control points more evenly spaced throughout the measurement domain?

Specific Comments: Pg. 1, Line 23 – Difficulty rather than difficult Pg 2, L8 – "area covered by imaged area" could be reworded. Pg. 2, L21 – be consistent with either 'freeze-up' or 'freeze up'. Pg. 2, L24 – control, rather than controls Pg. 2, L32 – unnecessary comma Pg. 3, L8 – Clarify the meaning of annual mean annual flow. Pg. 3, L21 – increase the number of images, rather than cameras? Pg. 4, L1 – I'd like to know more about what constitutes a quality index of 0.8. Pg. 4, L5 – 14 – There are four instances where the word 'are' or 'was' should be replaced with 'were'. Pg. 4, L27 - ...mechanisms that form ... Pg. 4, L32 – delete the word 'were' Pg. 5, L1 – the shear wall heights were ... Pg. 5, L5 – Figure rather than figure Pg. 6, L12 – snow cover develops Pg. 7, L7 – Though rather than thought Table 1 caption – control rather than Control. Caption Figure 3 – "shows and the two sections" should be reworded.

---

## Author Comment (AC2) · 6 Dec 2017

Dear reviewer, Please see the uploaded file "Response_to_review" that contains the authors response to all three reviews.

---

## Author Comment (AC3) · 6 Dec 2017

Dear reviewer, Please see the uploaded file "Response_to_review" that contains the authors response to all three reviews.
* * *

---

## Author Comment (AC4) · 6 Dec 2017

Dear reviewer, Please see the uploaded file "Response_to_review" that contains the authors response to all three reviews.
* * *

---

## Author Response (AR1)

**Response to the review of "Brief Communication: Mapping river ice using drones and structure from motion".**

The authors wish to thank the reviewers for their constructive comments and corrections to the discussion paper. In the following we have responded to each of the comments from each reviewer. The comment from the reviewer is in *italic font* while the response is in blue normal font.

First a general comment to all reviewers. From a comment from Mark Loewen on the description of the number of images used in the assessment in the Haga bru case, we noticed that we had used a dataset with a lower image quality treshold for the construction of the DEM. The results in the discussion papers are correct given the data used, but the Haga bru case is not done with the same level of image selection as the Sokna case and it does not follow the correct procedure according to the description in the paper. We therefore re-ran the Haga bru analysis to make the cases comparable. This lead to minor changes in Table 1, to Figure 1 and 2 and to the computed size of the ice jam on page 5, lines 5 – 10. We thank the reviewer for pointing us in the direction of this and we are sorry for overlooking this mistake in preparing the discussion paper.

**RC1: Responses to Karl-Erich Lindenschmidt**

*This communique highlights the application of drones in river ice work. Such applications are becoming more and more essential as advances in river ice research are requiring more detailed descriptions of the characteristics and geomorphological settings of ice, hence this note is timely. It introduces a method to other river ice researchers seeking an inexpensive and safe alternative for ice cover mapping.*
*Before this communique is accepted for publication, some amendments and revisions are required:*
*Limitations of the photography are extensively elucidated (particularly on Page 6), however little is said about the limitations of using UAVs themselves. Examples may include: - additional qualifications and registration required of the UAV operator, - operations feasible only on wind-calm days, - only short flights are possible (also through regulations) and - UAVs cover smaller aerial extents compared to other methods. Perhaps these can be listed in the Introduction so that the reader can assess if these limitations would hinder their particular case studies.*

We agree with the reviewer that this is important information for those that might be interested in employing a drone for assessment of ice. We have added a sentence in the introduction about limitations, and a more detailed paragraph in the discussion with some more information on regulations, flight time, flight distance and other factors related to the operation of a small drone.

In the introduction we added: "A possible drawback with the small drones as employed here is limits to flight distances (typically 1000-1300 meters) and battery life time in cold conditions."

In the discussion: "There are limitations to the application of the small drones in focus here. From experience, we expect practical flight range of about 1000 – 1300 meters in open terrain when flying in non-autonomous mode. This limit the application in large rivers, and makes it necessary to fly from several locations to cover longer reaches. The application of the small drone is also influenced by wind, and calm days are recommended both due to operation and for the best possible image quality. Another issue is related to battery life time in cold weather. The flight time is around 20 minutes, and it is necessary to pay attention to the battery status to avoid sudden loss of voltage in cold batteries. For sustained operation in cold climate several batteries are recommended, and spare batteries should be kept as warm as possible. A last issue to consider is operational constraints set by federal aviation regulations. In Europe, systems with a take-off weight of less than 2 kilos have less restrictions than larger systems but this must be checked with local regulations in each case"

*The section of consolidated ice shown in Figure 1 is labelled 'ice jam'. This may not be quite correct since ice jams usually extend across the river width to cause backwater staging. Perhaps labelling this section as 'consolidated ice' or 'ice-jam remnant' is be more accurate.*

The ice left on the bank is a part of an ice jam that covered the entire river for a period, but a part of it was removed by high water. We have updated the text to "ice-jam remnant" which we agree is a better description.

*Line 27 on Page 5 refers to "free water". I'm not sure what is meant here. "Open water" doesn't quite fit either since how can river ice have open water. Please restructure the sentence to clarify its meaning.*

We refer to an ice cover with sections of open water, so "free water" is replaced with "open water sections"

*Some minor, editorial revisions include:*
*Page 1, Line 23: replace "difficult and potentially dangerous" to "difficulties and dangers"*

We have removed "wrought with" as suggested by the second reviewer and "difficult and potentially dangerous" should therefore be ok.

*Page 1, Line 30: "ice processes dynamics" should read "ice process dynamics" or "the dynamics of ice processes".*
Updated
*Page 2, Line 33: perhaps replace "built" with "constructed"*
Updated
*Page 3, Line 30: "were", not "was"*
Updated
*Page 5, Line 17: "where", not "were"*
Updated

**RC2: Response to Mark Loewen**

*This is a timely and interesting paper describing an efficient and economical method for using inexpensive aerial drones to map river ice. I am quite certain that this method will prove to be a valuable new tool for river ice researchers. The method does not appear to be novel but its application to mapping of river ice is new and innovative. Accurate estimates of the aerial extent of various ice types can be made and under the right circumstances estimates of ice volumes are possible. The authors point out that the method allows for repetitive measurements to be made of the same reach because data can be acquired quickly and relatively easily. This will allow researchers to study the evolution of ice covers and the associated processes in much more detail than was previously possible.*

*The authors could consider discussing the following questions: How does the limited range of these inexpensive drones impact the applicability of this method to large rivers? I have deployed a similar drone on the Peace River, Canada and the width of the river exceeded the range in some places.*

We added a paragraph in the discussion on limitations using the small drone (also requested by the 1st reviewer). The practical flight distance of the drone when flying under control is an issue in larger rivers, and fixed wing drones may be a better option here.

For updates to the text, see the response to the first reviewer.

*In extremely cold weather battery life can drop by 50%. Did the authors encounter any difficulties because of this effect?*

This is an issue we have experienced when operating in cold weather, and care must be taken to avoid problems with a sudden drop in voltage. We added some text on this in the discussion and recommends careful attention to battery levels and to have several spare batteries stored as warm as possible when operating in cold climate.

In the discussion: "The flight time is around 20 minutes, and it is necessary to pay attention to the battery status to avoid sudden loss of voltage in cold batteries. For sustained operation in cold climate several batteries are recommended, and spare batteries should be kept as warm as possible"

*Would a higher resolution camera improve the accuracy of this method?*

Well yes there will be improvement. The important thing is the sensor size for the resolution and the capability of handling low light conditions. The resolution impacts the SfM workflow in principally two ways: 1) Feature matching between two cameras depends on multiscale image regions common to two images. Therefore, when the resolution is increased, the quality and quantity of features tends to increase. This improves the camera position and orientation estimates of the modelled regions. 2) Depth maps are needed to create the dense point cloud reconstruction. These maps are derived from ray-tracing based on the original images. If the images have higher spatial resolution, the quality of the depth maps, and therefore the dense clouds also tend to increase. Some text is added to the discussion on camera quality.

Updated paragraph in the discussion: "New versions of drones are appearing regularly, and improved sensor technology and light handling will improve picture quality and thereby the feature matching between cameras and the ability to generate better quality depth maps. This will increase the accuracy of the SfM generated DEM and orthophoto mosaics."

The paper is well written and the topic will be of interest to readers of the journal. I recommend publication after the authors have addressed the minor revisions suggested below.
Specific Comments:
Page 1-Line 23: Delete "wrought with".
Done.

Page 2- Line 29: ". . .and show examples of output of the method." Awkward wording.
The last part of the sentence is removed.

Section 2.1: Please provide more complete descriptions of the study reaches e.g. widths, slopes, geomorphology etc.
We have added some more information on the river reaches.

Added text: "The site in Gaula is about 75 meters wide and flows mainly over a bed of coarse gravel and smaller cobbles. At the site where the ice jam formed, the river narrows and the flow changes from a section with fast riffles to a deeper pool area. The Sokna site is steep (1/100) and consist of large cobbles and boulders and the river is characterised by short pools interspersed with drops and fast riffles/rapids. The width of the reach is around 18 meters."

Page 3-Lines 21-23: Unclear - there is only one camera on the drone so increasing the number of cameras is clearly done in the software. Please explain this more fully.
We agree that this was a somewhat confusing statement. We have upgraded this to rather explain how many pictures that was taken in total compared to the pictures we used in the SfM analysis. In doing this, we discovered that the reported data from Gaula was not the dataset prepared using the 0.8 image quality threshold. This is now upgraded and it led to a small adjustment of the results presented in table 1, figure 1 and figure 2 in addition to the computed volumes and ice depths. This slipped through the quality check in the original manuscript and we thank the reviewer for leading us on to this problem. We have added a comment to all reviewers at the start of the document to point to this change in the manuscript.

Changed text: "Of these, 82 pictures was used in the SfM analysis for Gaula (38%) and 68 pictures in Sokna (13%)."

Page 3-Lines 28-29: Are the 9 points referred to here called Control Points in Table 1? This is unclear.
Yes, they are now referred to as control points both in the text and in the table.

Page 4-Line 1: Should this read ". . . index greater than 0.8. . .". Seems odd that when a quality index is greater than 0.8 that images were excluded. If space permits a brief explanation of how this quality index is computed would be helpful.

We have clarified this in the text. It is not perfectly clear from the Agisoft documentation how the quality index is measured, but experiences we have is that the contrast distribution in the image is the variable measured in the quality index.

Added text: "The quality index is computed by Agisoft Photoscan, and even if the documentation is not very clear on this issue, experience shows that it is related to the spatial distribution of contrast in the images."

*Page 4-Lines18-21: The authors write that "...the digital elevation model is considered good. . .". They are referring to the errors listed in Table 1 but a brief discussion of how they arrived at this conclusion would be helpful to readers who are not familiar with DEM's.*
The text is upgraded to state that ice features can be found with high precision given the deviations found in the GCP and control point measurements.

*Page 4-Line 33: Delete "varied from".*
Corrected

*Page 5-Line 1: Change to ". . .were 1.12. . .".*
Corrected

*Page 5-Lines 2-5: The meaning of "outermost" and "outer part" are a bit unclear.*
We have tried to make this clearer by stating that it is from 25 meter

*Page 5-Line 22-23: Why was access to the Sokna reach difficult or impossible? Is it difficult to get to the stream or is it the ice conditions that make it unsafe?*
It was due to the ice conditions. The text is updated to state this.

*Page 5-Lines 30-33 & Page 6-Lines 1-3: It is not clear to me how determining the open water elevation helps to assess the thickness of anchor ice dams. Anchor ice dams are anchored either to the bed or large rocks so how is their thickness related to the open water elevation? Perhaps I am missing something here but the middle section of this paragraph confused me.*
At the study site, we could see the rocks that held the remnants of the broken anchor ice dams, and from these observations we could see that using the water surface as a basis for the computation of the thickness of the anchor ice dams would be reasonable. We do agree with the reviewer that this is not a generally applicable method and have updated the text to state this more clearly.

Adjusted text: "At the study site in Sokna, we could see that the open water would give a reasonable assessment of the thickness of the dams, but this do require field observations to confirm the method. In general, using the visible water surface has its limitations due to the turbulent surface of the water, and because the rocks and outcroppings known to be important for anchor ice dam formation may not be visible in the images captured after the ice is formed."

*Page 7=Lines 1-6: Much of this is repetitive. Perhaps it could be deleted?*
We do think that the comparison to other methods are relevant, and we are therefore a bit reluctant to remove this entirely. We have tried to rewrite the first part to make it less repetitive.

*Page 7-Lines 14-15: This is unclear, ". . .could benefit process understanding and model development and also the calibration and validation process." Please clarify.*
The sentence is reworded and improved.

*Figure 1: Unclear, is camera position the location of the drone when a single image was taken? What is the red arrow in the lower left corner? If it is flow direction, please relocate it and label it or refer to it in the caption. Note that flow direction should be indicated in all figures.*
The camera positions are the location of the drone for each picture, and the red arrow is the flood direction. The figure and caption is updated

*Figure 2: Label the plots A, B and C. Where are the last 10 m in the plots? Or perhaps the last sentence of the caption can be deleted since I think this information should be in the text.*
This is explained in the text and the sentence is deleted from the caption.

*Figure 3: The last sentence in the caption has some grammatical errors.*
The sentence is updated

**RC3: Response to Shawn Clark**

*The authors have presented an interesting paper regarding the use of unmanned aerial vehicles to map river ice extent and ice properties such as ice thickness and volume. It has also been our group's experience that UAVs can be very helpful for this purpose, as well as to quantify surface ice concentration and ice pan velocities. Their methodology should be useful for many other researchers in the field as well. I have a few minor comments, followed by some grammatical suggestions. It would have been helpful to have additional detail regarding the specifics of what constitutes a high quality image vs a low quality image, and how one might be more likely to achieve the former. Any practical tips for a successful flight would be appreciated by the readers.*
We have added a paragraph in the discussion (as also requested by the other reviewers) with some information on flight times, battery issues, flight distances and flight regulations. We have also added some more information on picture quality and issues related to light which is important for good quality in images and the following SfM analysis.

We have added some more info on this in the discussion, see response to the first reviewer for the specific text.

*It also would have been nice to see whether the chosen number of control points was actually necessary. Is it a coincidence that the number of points only varied between 9 and 11? Would the error have been significantly increased if only 5 control points had been used? Are there any suggestions for the placement of these points? For instance, had the opposite bank been easily accessible, would it have been better to have the control points more evenly spaced throughout the measurement domain?*

The number of points are mainly based on experience from previous applications of the SfM method. Goldstein et al. (2015) shows that quality of the georeferencing increases when points are increased up to 10, but adding more points above 10 give little improvement. We will add a comment on this in the manuscript.

It is important to spread the control points in the x and y direction and to avoid to have them in a "straight" line. For Sokna it would have been good to have points on both sides of the river, but the opposite bank was out of reach at the day of measurement.
Addition to text: "The number of GCPs is based on previous experience, and the number corresponds with the findings of Goldstein et al. (2015) who showed increased quality when the number of GCPs were increased up to 10 and little improvement when more points were added."

Goldstein, EB, Oliver, AR, deVries, E, Moore, LJ and Jass, T. (2015) Ground control point requirements for structure-from-motion derived topography in low- slope coastal environments. PeerJ PrePrints | https://dx.doi.org/10.7287/peerj.preprints.1444v1

*Specific Comments:*
*Pg. 1, Line 23 – Difficulty rather than difficult*
The sentence is updated (also commented by the two other reviewers)
*Pg 2, L8 – "area covered by imaged area" could be reworded.*
This is reworded
*Pg. 2, L21 – be consistent with either 'freeze-up' or 'freeze up'.*
Updated
*Pg. 2, L24 – control, rather than controls*
Updated
*Pg. 2, L32 – unnecessary comma*
Updated
*Pg. 3, L8 – Clarify the meaning of annual mean annual flow.*
This was an error, now updated to "28% of the mean annual flow"
*Pg. 3, L21 – increase the number of images, rather than cameras?*
This is also pointed out by reviewer2, and the sentence is reworded to be clearer and to remove a possible misunderstanding between images and cameras. See comment to all reviewers at the start of the response document.
*Pg. 4, L1 – I'd like to know more about what constitutes a quality index of 0.8.*
It is not perefectly clear from the Agisoft documentation how the quality index is measured, but experiences we have is that the contrast distribution in the image is the variable measured in the quality index. We have added some info in the text.
*Pg. 4, L5 – 14 – There are four instances where the word 'are' or 'was' should be replaced with 'were'.*
Updated
*Pg. 4, L27 - . . .mechanisms that form . . .*
Updated
*Pg. 4, L32 – delete the word 'were'*
Sentence is updated
*Pg. 5, L1 – the shear wall heights were . . .*
Updated

*Pg. 5, L5 – Figure rather than figure*
Updated
*Pg. 6, L12 – snow cover develops*
Updated
*Pg. 7, L7 – Though rather than thought*
Updated
*Table 1 caption – control rather than Control.*
Updated
*Caption Figure 3 – "shows and the two sections" should be reworded.*
Updated

[revised manuscript text omitted]

---

## Editor Decision (ED1)

Dear Dr. Alfredsen,

Thank you for your revised manuscript. It will be suitable for publication in TC with some minor revisions that I have suggested below.

Best regards,

Peter

Editorial comments:

P1,L10: "supply, riparian" becomes "supply, and riparian"

P1,L13: "and satellite imagery, data from satellite-based radars as increasingly aerial and terrestrial imagery are currently applied" becomes "and data from satellite-based sensors and, increasingly, aerial and terrestrial imagery are currently applied"

P1, L17: "the ice and with considerable less measurement efforts compared to traditional surveying methods" becomes "the ice, and with considerably less measurement effort compared to traditional surveying methods.

P1, L17: "The methodology applied to ice mapping is outlined here, and examples are shown on how to successfully apply the method to derive data on ice processes" becomes "A methodology applied to ice mapping is outlined here, and examples are shown on how to successfully derive quantitative data on ice processes."

P1, L22: "Physical monitoring" becomes "Physically monitoring"

P1, L25: "integrated satellite images (MODIS) with radar data (RADARSAT-2)" becomes "integrated optical (MODIS) and radar (RADARSAT-2) satellite data"

P1, L26: "also evaluated the dataset with aerial- and terrestrial imagery to evaluate the satellite" becomes "also used aerial- and terrestrial imagery to validate the satellite". Please check this edit to see if it is true.

P1, L28: "successive satellite imagery" becomes "successive radar? optical? satellite images". Did they use optical or radar, or both? Please clarify.

P2, L1-5: "time series of these ice data for the lower Nelson River, Canada. Most satellite-based approaches described in the existing literature are applied to large rivers and may not be applicable to smaller rivers and streams due to the resolution of the satellite imagery. Methods are therefore needed to monitor ice formation in smaller rivers. Further work is also needed to

move from a qualitative evaluation of ice (focussing on ice types and presence or non-presence of ice) towards a quantification of ice volumes and the spatial distribution 5 of ice" becomes "time series of these ice data for the lower Nelson River, Canada. Most satellite-based approaches described in the literature are applied to large rivers and may not apply to smaller rivers and streams due to the coarse resolution of the satellite imagery. Remote sensing methods are therefore needed to monitor ice formation in smaller rivers. Further work is also needed to move from a qualitative evaluation of ice (focussing on ice types and presence or non-presence of ice) towards a quantification of ice volumes and the spatial distribution."

P2, L6: "aerial drones. Such drones increasingly have camera systems of sufficient quality, and the resultant aerial" becomes "aerial drones with camera systems of sufficient quality that the resultant aerial"

P2, L9: "Combined with ground control points, surface models can be georeferenced and combined with other spatial data for analysis. Based on the georeferenced point cloud and aerial images, digital elevation models of high accuracy and resolution can be developed" becomes "Ground control points can be used to georeference the point cloud and aerial images generated by the drone, in order to develop digital elevation models of high accuracy and resolution.

P2, L11: "Additionally, derived surface texture can be overlaid onto surface models and imagery can be undistorted and reprojected as georeferenced orthophoto mosaics to provide an accurate aerial image of the study object" becomes "Additionally, surface texture may be derived and overlaid on the surface models, and derived georeferenced orthophoto mosaics to provide an accurate aerial image of the study object"

P2, L14: "needed (Westoby et al., 2012;Smith et al., 2016), where the method" becomes "needed (Westoby et al., 2012; Smith et al., 2016), and the method"

P2, L16: "(Vasquez-Tarrio et al., 2017;Arif et al., 2016)" Becomes "(Vasquez-Tarrio et al., 2017; Arif et al., 2016).

P2, L17: "glaciers e.g (Ryan et al., 2015) and in the study of snow accumulation (Nolan et al., 2015)" becomes "glaciers (e.g. Ryan et al., 2015) and snow accumulation (e.g. Nolan et al., 2015)."

P2, L20: "jams forming" becomes "jams formed"

P2, L21: "dams forming" becomes "dams formed"

P2, L23-26: "flooding and thereby severe damage to infrastructure and the riverine flora and fauna (Beltaos, 1995;Prowse and Culp, 2003). Anchor ice dams control the freeze–up process in small rivers and streams, and are thereby important for understanding winter 25 conditions in such streams (Stickler et al., 2010;Turcotte and Morse, 2011). A possible drawback with the small drones as employed here is limits to flight distances (typically 1000-1300 meters) and

battery life time in cold conditions" becomes "flooding, and thereby severe damage to infrastructure and the riverine flora and fauna (Beltaos, 1995; Prowse and Culp, 2003). Anchor ice dams control the freeze–up process in small rivers and streams, and are thus important for understanding winter 25 conditions in such streams (Stickler et al., 2010; Turcotte and Morse, 2011). A possible drawback with the use of small drones as employed here is limited flight distances (typically 1000-1300 m) and battery life time in cold conditions"

P2, 29-30: "Furthermore, we use the method to map the remnant of a stranded ice jam and a river section 30 with anchor ice dams." becomes "Furthermore, we use the method to map the remnant of a stranded ice jam in one river section another river section with anchor ice dams"

P2, L33: "quality of data available and further improve our" becomes "quality of data available. This may further improve our"

P3, L6:" Haga bru and" becomes "Haga bru, and"

P3, L7: "The length" becomes "The lengths"

P3, L8-13: "are 350 m and 200 m for Gaula and Sokna respectively. During the measurements, the discharge in Gaula (at gauge Gaulfoss) was approximately 20 $m^3s^{-1}$ and in Sokna (at gauge Hugdal bru) approximately 3.5 $m^3s^{-1}$. This correspond to 28% of the mean 10 annual flow. The site in Gaula is about 75 meters wide and flows mainly over a bed of coarse gravel and smaller cobbles. At the site where the ice jam formed, the river narrows and the flow changes from a section with fast riffles to a deeper pool area. The Sokna site is steep (1/100) and consist of large cobbles and boulders and the river is characterised by short pools interspersed with drops and fast riffles/rapids. The width of the reach is around 18 meters" becomes "are 350 m and 200 m for Gaula and Sokna, respectively. During the measurements, the discharge in Gaula (at the Gaulfoss gauge) was approximately 20 $m^3$ $s^{-1}$ and in Sokna (at the Hugdal bru gauge) approximately 3.5 $m^3$ $s^{-1}$. These rates correspond to 28% of the mean annual flow. The Gaula site is about 75 m wide, and water flows mainly over a bed of coarse gravel and smaller cobbles. At the location where the ice jam formed, the river narrows and the flow changes from a section with fast riffles to a deeper pool area. The Sokna site is steep (1/100), the bed consist of large cobbles and boulders, and the river is characterised by short pools interspersed with drops and fast riffles/rapids. The width of the reach is around 18 m"

P3, L21-22: "Apple iPad with the DJI Go application for flight control. Though capable of autonomous operation according to a pre-planned flight plan with the DJI GS Pro application" becomes "Apple iPad with the DJI Go application. Though capable of autonomous operation according to a pre-planned flight plan generated with the DJI GS Pro application"

P3, L24: "information and the GPS" becomes "information and GPS"

P3, L25: "82 pictures was" becomes "82 pictures were"

P3, L27: "and 10 minutes was used for the flight" becomes "and the flight lasted 10 minutes"

P3, L29: "and they were identified using" becomes "and they consisted of"

P3, L30: After the sentence on the markers, and before the note on Gaula site GCPs, insert the following (or similar): "GCPs were spread out over the measurement domains as much as conditions allowed, but only one bank was accessible to us at the Sokna site."

P4, 2-4: What did you learn from your previous experience? Perhaps you can expand you your experience? I ask because you cannot use Goldstein et al (2015) as a reference. Their document is not peer-reviewed, and remains un-reviewed this many years later. Please delete from your reference list.

P4, L8: "computed by Agisoft Photoscan, and even if the documentation is not very clear on this issue, experience". Please contact the software company and clarify the issue here for all readers. Then you can say "computed by Agisoft Photoscan based on…, and experience shows"

P4, L14: "Each marker were identified" becomes "Each marker was identified"

P4, L19: "the DEM and an orthophoto" becomes "the surface model, and an orthophoto". Throughout the text I suggest calling your product the surface model rather than the "DEM" to distinguish it from the Norwegian DEM that you also use.

P4, Lines 20-22: Replace "was" with "were" in both instances. Change "in Gaula were taken from the 1 meter" to "in Gaula was taken from the 1 m"

P4, L26: "computed in Photoscan, and the results are shown in Table 1. The accuracy of the digital elevation model is considered good considering the errors in table 1, and ice features can be derived with high precision" becomes "computed in Photoscan. Given the low errors shown in Table 1, the accuracy of the digital elevation model is considered good and ice features may be derived with high precision"

P4, L31: "Gaula site, and the extent of the ice jam and the directional" becomes "Gaula site, the extent of the ice jam, and the directional"

P5, L5: "of the 5 SfM- derived DEM and" becomes "of the 5 SfM-derived DEM and"

P5, L6: "ice event and from observations in the field the jam" becomes "ice event, and from observations in the field, the jam

P5, L8-9: Transects A, B, and C should be A-A', B-B', and C-C'.. Please change in the text, and modify Figure 2 accordingly. This will make it obvious where the start and end of each transect is. With the orientation known, some text later on in this paragraph will make more sense.

P5, L10-14: "This is most likely a minimum volume since it is difficult to determine if the outermost meters of the DEM describing the ground is the actual ground level or the water surface of the river. The DEM used does not cover terrain under water, and it is not clear if we see the bottom or the water surface in the outer part of the ground level plot on Figure 2" becomes "This is most likely a minimum volume since it is probable that the Norwegian national DEM describes the river water surface rather than actual ground level, in particular for the last 10 m of each transect (Figure 2)."

P5, L26: "measured ground points the point cloud was georeferenced, and a digital elevation model and a georeferenced orthophoto mosaic developed" becomes "measured ground points, the point cloud was georeferenced and a digital elevation model and a georeferenced orthophoto mosaic were developed"

P5, L27-28: "data on ice thickness, volume and spatial distribution was extracted" becomes "data on ice thickness, volume, and spatial distribution were extracted"

P5, L29: "(e.g. Timalsina (2014))," becomes "(e.g. Timalsina, 2014),

P5, L31: "was possible our new" becomes "was possible, our new"

P6, L2-10: "As is in the case with the Gaula ice jam, it is relatively simple to compute ice volumes and derive ice cross sections for ice grounded on a surface with known ice-free geometry. In cases where river ice has open water sections, the problem is more complicated, as is illustrated by considering the conditions shown in the Sokna case (Figure 3). Anchor ice dam positions and their widths and heights can be derived from the picture and the DEM, but unlike the Gaula case, no ice-free geometry of the river is available to assess the volume of ice. A possible approach as we show is to derive the elevation of the level of the open water from the SfM generated DEM and then use this to assess the thickness of the anchor ice dams seen in the picture. At the study site in Sokna, we could see that the open water would give a reasonable assessment of the thickness of the dams, but this do require field observations to confirm the method" becomes "At the Gaula ice jam, it was relatively simple to compute ice volumes and derive ice cross sections for ice grounded on a surface with known ice-free geometry, but in cases where river ice has open water sections the problem is more complicated. This is illustrated by considering the conditions shown in the Sokna case (Figure 3). Anchor ice dam positions and their widths and heights can be derived from the orthophoto mosaic and the surface model, but unlike the Gaula case, no ice-free geometry of the river was available to assess the volume of ice. A possible approach as we show is to derive the elevation of the level of the open water from the SfM generated surface model and then use this to assess the thickness of the anchor ice dams seen in the mosaic. At the Sokna site, we could see that the open water would give a reasonable assessment of the thickness of the dams, but this approach requires field observations to confirm the method"

P6, L25: Join to end of pervious paragraph. "6.17/55 mm" becomes "6.17 × 55 mm"

P6, L26: Period after ([www.dji.com](www.dji.com))

P7, L2: "improve picture quality and thereby the feature matching between cameras and the ability to generate better quality depth maps" becomes "improve picture quality, feature matching between cameras, and depth maps"

L7, L13-21: Much of this is repetitive, and similar to text on P5, L21-26. Please integrate the text here (or there).

P7, L23: " the small drones in focus here" becomes "the DJI Phantom 3 Professional used here"

P7, L24: "1000 – 1300 meters" becomes "1000 – 1300 m"

P7, L26: "Another issue is related to" becomes "Another issues relates to"

P7, L29-31: "A last issue to consider is operational constraints set by federal aviation regulations. In Europe, systems with a take-off weight of less than 2 kilos have less restrictions than larger systems but this must be checked with local regulations in each case." Becomes "A last issue to consider are operational constraints set by applicable aviation regulations that must be consulted before each flight, which may limit system size, mass, range, elevation, timing, etc."

P8, L1: " data of the" becomes "data on the"

P8, L2: "variables the DEM and" becomes "variables, the surface model and"

P8, L3: "Data can also" becomes "Data generated from SfM can also"

P8, L31: Delete Goldstein et al., 2015.

Figure 1: Red crosses are very hard to see. Please make bigger. You could also enlarge the blue triangles. The crosses on the railroad are also difficult to see. Please increase the thickness of the railroad line.

Figure 2: Labels for transects A, B, and C are cut off. Please change transect names to A-A', B-B', and C-C'.

---

## Author Response (AR2)

Dear Editor,

Thank you for the comments on the manuscript. Below you will find the responses to the comments. Comments in black normal font, responses in blue italic font. A mark-up version of the manuscript is also provided.

Best Regards,
Knut Alfredsen

Dear Dr. Alfredsen,

Thank you for your revised manuscript. It will be suitable for publication in TC with some minor revisions that I have suggested below.

Best regards, Peter

Editorial comments:

P1,L10: "supply, riparian" becomes "supply, and riparian"

*Corrected*

P1,L13: "and satellite imagery, data from satellite-based radars as increasingly aerial and terrestrial imagery are currently applied" becomes "and data from satellite-based sensors and, increasingly, aerial and terrestrial imagery are currently applied"

*Corrected*

P1, L17: "the ice and with considerable less measurement efforts compared to traditional surveying methods" becomes "the ice, and with considerably less measurement effort compared to traditional surveying methods.

*Corrected*

P1, L17: "The methodology applied to ice mapping is outlined here, and examples are shown on how to successfully apply the method to derive data on ice processes" becomes "A methodology applied to ice mapping is outlined here, and examples are shown on how to successfully derive quantitative data on ice processes."

*Corrected*

P1, L22: "Physical monitoring" becomes "Physically monitoring"

*Corrected*

P1, L25: "integrated satellite images (MODIS) with radar data (RADARSAT-2)" becomes "integrated optical (MODIS) and radar (RADARSAT-2) satellite data"

*Corrected*

P1, L26: "also evaluated the dataset with aerial- and terrestrial imagery to evaluate the satellite" becomes "also used aerial- and terrestrial imagery to validate the satellite". Please check this edit to see if it is true.

*Corrected. This is correct, time-lapse and aerial images was used to verify the satellite.*

P1, L28: "successive satellite imagery" becomes "successive radar? optical? satellite images". Did they use optical or radar, or both? Please clarify.
*They used successive stereo images from the Japanese ALOS satellite. The text now specifies "images"*

P2, L1-5: "time series of these ice data for the lower Nelson River, Canada. Most satellite-based approaches described in the existing literature are applied to large rivers and may not be applicable to smaller rivers and streams due to the resolution of the satellite imagery. Methods are therefore needed to monitor ice formation in smaller rivers. Further work is also needed to move from a qualitative evaluation of ice (focussing on ice types and presence or non-presence of ice) towards a quantification of ice volumes and the spatial distribution 5 of ice" becomes "time series of these ice data for the lower Nelson River, Canada. Most satellite-based approaches described in the literature are applied to large rivers and may not apply to smaller rivers and streams due to the coarse resolution of the satellite imagery. Remote sensing methods are therefore needed to monitor ice formation in smaller rivers. Further work is also needed to move from a qualitative evaluation of ice (focussing on ice types and presence or non-presence of ice) towards a quantification of ice volumes and the spatial distribution."
*Corrected*

P2, L6: "aerial drones. Such drones increasingly have camera systems of sufficient quality, and the resultant aerial" becomes "aerial drones with camera systems of sufficient quality that the resultant aerial"
*Corrected*

P2, L9: "Combined with ground control points, surface models can be georeferenced and combined with other spatial data for analysis. Based on the georeferenced point cloud and aerial images, digital elevation models of high accuracy and resolution can be developed" becomes "Ground control points can be used to georeference the point cloud and aerial images generated by the drone, in order to develop digital elevation models of high accuracy and resolution.
*Corrected*

P2, L11: "Additionally, derived surface texture can be overlaid onto surface models and imagery can be undistorted and reprojected as georeferenced orthophoto mosaics to provide an accurate aerial image of the study object" becomes "Additionally, surface texture may be derived and overlaid on the surface models, and derived georeferenced orthophoto mosaics to provide an accurate aerial image of the study object"
*Corrected*

P2, L14: "needed (Westoby et al., 2012;Smith et al., 2016), where the method" becomes "needed (Westoby et al., 2012; Smith et al., 2016), and the method"
*Corrected*

P2, L16: "(Vasquez-Tarrio et al., 2017;Arif et al., 2016)" Becomes "(Vasquez-Tarrio et al., 2017; Arif et al., 2016).

*Corrected*

P2, L17: "glaciers e.g (Ryan et al., 2015) and in the study of snow accumulation (Nolan et al., 2015)" becomes "glaciers (e.g. Ryan et al., 2015) and snow accumulation (e.g. Nolan et al., 2015)."

*Corrected*

P2, L20: "jams forming" becomes "jams formed" P2, L21: "dams forming" becomes "dams formed"

*Corrected*

P2, L23-26: "flooding and thereby severe damage to infrastructure and the riverine flora and fauna (Beltaos, 1995;Prowse and Culp, 2003). Anchor ice dams control the freeze–up process in small rivers and streams, and are thereby important for understanding winter 25 conditions in such streams (Stickler et al., 2010;Turcotte and Morse, 2011). A possible drawback with the small drones as employed here is limits to flight distances (typically 1000-1300 meters) and battery life time in cold conditions" becomes "flooding, and thereby severe damage to infrastructure and the riverine flora and fauna (Beltaos, 1995; Prowse and Culp, 2003). Anchor ice dams control the freeze–up process in small rivers and streams, and are thus important for understanding winter 25 conditions in such streams (Stickler et al., 2010; Turcotte and Morse, 2011). A possible drawback with the use of small drones as employed here is limited flight distances (typically 1000-1300 m) and battery life time in cold conditions"

*Corrected*

P2, 29-30: "Furthermore, we use the method to map the remnant of a stranded ice jam and a river section 30 with anchor ice dams." becomes "Furthermore, we use the method to map the remnant of a stranded ice jam in one river section another river section with anchor ice dams"

*Corrected*

P2, L33: "quality of data available and further improve our" becomes "quality of data available. This may further improve our"

*Corrected*

P3, L6:" Haga bru and" becomes "Haga bru, and"

*Corrected*

P3, L7: "The length" becomes "The lengths"

*Corrected*

P3, L8-13: "are 350 m and 200 m for Gaula and Sokna respectively. During the measurements, the discharge in Gaula (at gauge Gaulfoss) was approximately 20 m$^3$s$^{-1}$ and in Sokna (at gauge

Hugdal bru) approximately 3.5 m$^3$s$^{-1}$. This correspond to 28% of the mean 10 annual flow. The site in Gaula is about 75 meters wide and flows mainly over a bed of coarse gravel and smaller cobbles. At the site where the ice jam formed, the river narrows and the flow changes from a section with fast riffles to a deeper pool area. The Sokna site is steep (1/100) and consist of large cobbles and boulders and the river is characterised by short pools interspersed with drops and fast riffles/rapids. The width of the reach is around 18 meters" becomes "are 350 m and 200 m for Gaula and Sokna, respectively. During the measurements, the discharge in Gaula (at the Gaulfoss gauge) was approximately 20 m$^3$ s$^{-1}$ and in Sokna (at the Hugdal bru gauge) approximately 3.5 m$^3$ s$^{-1}$. These rates correspond to 28% of the mean annual flow. The Gaula site is about 75 m wide, and water flows mainly over a bed of coarse gravel and smaller cobbles. At the location where the ice jam formed, the river narrows and the flow changes from a section with fast riffles to a deeper pool area. The Sokna site is steep (1/100), the bed consist of large cobbles and boulders, and the river is characterised by short pools interspersed with drops and fast riffles/rapids. The width of the reach is around 18 m"
*Corrected*

P3, L21-22: "Apple iPad with the DJI Go application for flight control. Though capable of autonomous operation according to a pre-planned flight plan with the DJI GS Pro application" becomes "Apple iPad with the DJI Go application. Though capable of autonomous operation according to a pre-planned flight plan generated with the DJI GS Pro application"
*Corrected*

P3, L24: "information and the GPS" becomes "information and GPS"
*Corrected*

P3, L25: "82 pictures was" becomes "82 pictures were"
*Corrected*

P3, L27: "and 10 minutes was used for the flight" becomes "and the flight lasted 10 minutes"
*Corrected*

P3, L29: "and they were identified using" becomes "and they consisted of"
*Corrected*

P3, L30: After the sentence on the markers, and before the note on Gaula site GCPs, insert the following (or similar): "GCPs were spread out over the measurement domains as much as conditions allowed, but only one bank was accessible to us at the Sokna site."
*Done*

P4, 2-4: What did you learn from your previous experience? Perhaps you can expand you your experience? I ask because you cannot use Goldstein et al (2015) as a reference. Their document

is not peer-reviewed, and remains un-reviewed this many years later. Please delete from your reference list.

*I added some words on previous experience based on Christian Haas and Jeffrey Tuthans work on SfM and the number and placement of GCPs. The Goldstein reference is removed and replaced by a reference to Turner et al (2012) that also shows that around 10 points provide good accuracy.*

*Turner, D., Lucier, A., and Watson, C.: An Automated Technique for Generating Georectified Mosaics from Ultra-High Resolution Unmanned Aerial Vehicle (UAV) Imagery, Based on Structure from Motion (SfM) Point Clouds, Remote Sensing, 4, 1392-1410,*

P4, L8: "computed by Agisoft Photoscan, and even if the documentation is not very clear on this issue, experience". Please contact the software company and clarify the issue here for all readers. Then you can say "computed by Agisoft Photoscan based on..., and experience shows"
*This is now updated based on info from Agisoft.*

P4, L14: "Each marker were identified" becomes "Each marker was identified"
*Corrected.*

P4, L19: "the DEM and an orthophoto" becomes "the surface model, and an orthophoto". Throughout the text I suggest calling your product the surface model rather than the "DEM" to distinguish it from the Norwegian DEM that you also use.
*"Surface model" is now used for the SfM generated model and DEM is used for the Norwegian national model through the manuscript.*

P4, Lines 20-22: Replace "was" with "were" in both instances. Change "in Gaula were taken from the 1 meter" to "in Gaula was taken from the 1 m"
*Corrected.*

P4, L26: "computed in Photoscan, and the results are shown in Table 1. The accuracy of the digital elevation model is considered good considering the errors in table 1, and ice features can be derived with high precision" becomes "computed in Photoscan. Given the low errors shown in Table 1, the accuracy of the digital elevation model is considered good and ice features may be derived with high precision"
*Corrected.*

P4, L31: "Gaula site, and the extent of the ice jam and the directional" becomes "Gaula site, the extent of the ice jam, and the directional"
*Corrected.*

P5, L5: "of the 5 SfM- derived DEM and" becomes "of the 5 SfM-derived DEM and"
*Corrected.*

P5, L6: "ice event and from observations in the field the jam" becomes "ice event, and from observations in the field, the jam
*Corrected.*

P5, L8-9: Transects A, B, and C should be A-A', B-B', and C-C'.. Please change in the text, and modify Figure 2 accordingly. This will make it obvious where the start and end of each transect is. With the orientation known, some text later on in this paragraph will make more sense.
*Corrected.*

P5, L10-14: "This is most likely a minimum volume since it is difficult to determine if the outermost meters of the DEM describing the ground is the actual ground level or the water surface of the river. The DEM used does not cover terrain under water, and it is not clear if we see the bottom or the water surface in the outer part of the ground level plot on Figure 2" becomes "This is most likely a minimum volume since it is probable that the Norwegian national DEM describes the river water surface rather than actual ground level, in particular for the last 10 m of each transect (Figure 2)."
*Corrected.*

P5, L26: "measured ground points the point cloud was georeferenced, and a digital elevation model and a georeferenced orthophoto mosaic developed" becomes "measured ground points, the point cloud was georeferenced and a digital elevation model and a georeferenced orthophoto mosaic were developed"
*Corrected.*

P5, L27-28: "data on ice thickness, volume and spatial distribution was extracted" becomes "data on ice thickness, volume, and spatial distribution were extracted"
*Corrected.*

P5, L29: "(e.g. Timalsina (2014)),"becomes "(e.g. Timalsina, 2014),
*Corrected.*

P5, L31: "was possible our new" becomes "was possible, our new"
*Corrected.*

P6, L2-10: "As is in the case with the Gaula ice jam, it is relatively simple to compute ice volumes and derive ice cross sections for ice grounded on a surface with known ice-free geometry. In cases where river ice has open water sections, the problem is more complicated, as is illustrated by considering the conditions shown in the Sokna case (Figure 3). Anchor ice dam positions and their widths and heights can be derived from the picture and the DEM, but unlike the Gaula case, no ice-free geometry of the river is available to assess the volume of ice. A possible approach as we show is to derive the elevation of the level of the open water from the SfM generated DEM and then use this to assess the thickness of the anchor ice dams seen in the picture. At the study site in Sokna, we could see that the open water would give a reasonable assessment of the thickness of the dams, but this do require field observations to confirm the

method" becomes "At the Gaula ice jam, it was relatively simple to compute ice volumes and derive ice cross sections for ice grounded on a surface with known ice-free geometry, but in cases where river ice has open water sections the problem is more complicated. This is illustrated by considering the conditions shown in the Sokna case (Figure 3). Anchor ice dam positions and their widths and heights can be derived from the orthophoto mosaic and the surface model, but unlike the Gaula case, no ice-free geometry of the river was available to assess the volume of ice. A possible approach as we show is to derive the elevation of the level of the open water from the SfM generated surface model and then use this to assess the thickness of the anchor ice dams seen in the mosaic. At the Sokna site, we could see that the open water would give a reasonable assessment of the thickness of the dams, but this approach requires field observations to confirm the method"
*Corrected.*

P6, L25: Join to end of pervious paragraph. "6.17/55 mm" becomes "6.17 ✎55 mm"
*The paragraphs are joined. I have changed 6.17/55 to 6.17 x 55 mm.*

P6, L26: Period after (www.dji.com)
*Corrected.*

P7, L2: "improve picture quality and thereby the feature matching between cameras and the ability to generate better quality depth maps" becomes "improve picture quality, feature matching between cameras, and depth maps"
*Corrected.*

L7, L13-21: Much of this is repetitive, and similar to text on P5, L21-26. Please integrate the text here (or there).
*Most of the text on page 5 is removed and the comparison is left on page 7.*

P7, L23: " the small drones in focus here" becomes "the DJI Phantom 3 Professional used here"
*Corrected.*

P7, L24: "1000 – 1300 meters" becomes "1000 – 1300 m"
*Corrected.*

P7, L26: "Another issue is related to" becomes "Another issues relates to"
*Corrected.*

P7, L29-31: "A last issue to consider is operational constraints set by federal aviation regulations. In Europe, systems with a take-off weight of less than 2 kilos have less restrictions than larger systems but this must be checked with local regulations in each case." Becomes "A last issue to consider are operational constraints set by applicable aviation regulations that must be consulted before each flight, which may limit system size, mass, range, elevation, timing, etc."
*Corrected.*

P8, L1: " data of the" becomes "data on the"
*Corrected.*

P8, L2: "variables the DEM and" becomes "variables, the surface model and"
*Corrected.*

P8, L3: "Data can also" becomes "Data generated from SfM can also"
*Corrected.*

P8, L31: Delete Goldstein et al., 2015.
*Done*

Figure 1: Red crosses are very hard to see. Please make bigger. You could also enlarge the blue triangles. The crosses on the railroad are also difficult to see. Please increase the thickness of the railroad line.
*The symbols are increased in size and figure 1 is updated.*

Figure 2: Labels for transects A, B, and C are cut off. Please change transect names to A-A', B- B', and C-C'.
*Figure is updated*

[revised manuscript text omitted]

---

## Editor Decision (ED2)

Dear Dr. Alfredsen,

Thank you for your revised manuscript. Please make the following technical corrections and the manuscript can proceed through publication.

Best regards,

Peter

P1, L4: Delete the second period at the end of the sentence.

P1, L14: Again, "(Vasquez-Tarrio et al.; 2017;Arif et al., 2016)" Becomes "(Vasquez-Tarrio et al., 2017; Arif et al., 2016). The semi-colon after "Vasquez-Tarrio et al." should be a comma, and insert a space before "Arif".

P1, L21: Insert a space before Prowse

P1, L23: insert a space before Turcotte

P3, L10: Change "smaller" to "small"

P7, L19: Change "issues" to "issue"

P8, L21: Capitalize the conference title. Please verify that the conference has been cited in the style expected in TC.

P8, L25: Delete the Goldstein reference.

P8, L40: Capitalize the journal title

---

## Author Response (AR3)

Dear Editor,

Thank you again for the comments to the manuscript. Below you will find the responses and a track changes version of the manuscript.

Best Regards,

Knut Alfredsen

Dear Dr. Alfredsen,
Thank you for your revised manuscript. Please make the following technical corrections and the manuscript can proceed through publication.
Best regards, Peter

P1, L4: Delete the second period at the end of the sentence.
*Deleted.*

P1, L14: Again, "(Vasquez-Tarrio et al.; 2017;Arif et al., 2016)" Becomes "(Vasquez-Tarrio et al., 2017; Arif et al., 2016). The semi-colon after "Vasquez-Tarrio et al." should be a comma, and insert a space before "Arif".
*Fixed, sorry for this.*

P1, L21: Insert a space before Prowse
*Fixed*

P1, L23: insert a space before Turcotte
*Fixed*

P3, L10: Change "smaller" to "small"
*Fixed*

P7, L19: Change "issues" to "issue"
*Fixed*

P8, L21: Capitalize the conference title. Please verify that the conference has been cited in the style expected in TC.
*Conference title is capitalized. I have also updated the reference to properly refer to the Proceedings of the conference and the page number of the article.*

P8, L25: Delete the Goldstein reference.
*Deleted again, I am not sure why it came back in the submitted version.*

P8, L40: Capitalize the journal title
*Fixed*

[revised manuscript text omitted]